# Fracture Characteristics of Commercial PEEK Dental Crowns: Combining the Effects of Aging Time and TiO_2_ Content

**DOI:** 10.3390/polym15122720

**Published:** 2023-06-17

**Authors:** Wen-Ju Lu, Wei-Cheng Chen, Viritpon Srimaneepong, Chiang-Sang Chen, Chang-Hung Huang, Hui-Ching Lin, Oi-Hong Tung, Her-Hsiung Huang

**Affiliations:** 1Department of Dentistry, National Yang Ming Chiao Tung University, Taipei 112, Taiwan; luanne30813002.de08@nycu.edu.tw (W.-J.L.); huangmmh.b533@mmh.org.tw (C.-H.H.); das49@tpech.gov.tw (H.-C.L.); tohong@vghtpe.gov.tw (O.-H.T.); 2Department of Medicine, MacKay Medical College, New Taipei City 252, Taiwan; wchena@mmc.edu.tw; 3Division of Sports Medicine & Surgery, Department of Orthopedic Surgery, MacKay Memorial Hospital, Taipei 104, Taiwan; 4Department of Prosthodontics, Faculty of Dentistry, Chulalongkorn University, Bangkok 10330, Thailand; viritpon.s@chula.ac.th; 5Department of Orthopedic Surgery, Far Eastern Memorial Hospital, New Taipei City 220, Taiwan; ccs0102@femh.org.tw; 6Department of Materials and Textiles, Asia Eastern University of Science and Technology, New Taipei City 220, Taiwan; 7Department of Medical Research, MacKay Memorial Hospital, Taipei 251, Taiwan; 8Department of Stomatology, Taipei City Hospital, Zhongxing Branch, Taipei 103, Taiwan; 9General Education Center, University of Taipei, Taipei 106, Taiwan; 10Department of Stomatology, Taipei Veterans General Hospital, Taipei 112, Taiwan; 11Institute of Oral Biology, National Yang Ming Chiao Tung University, Taipei 112, Taiwan; 12Department of Medical Research, China Medical University Hospital, China Medical University, Taichung 404, Taiwan; 13Department of Bioinformatics and Medical Engineering, Asia University, Taichung 413, Taiwan; 14School of Dentistry, Kaohsiung Medical University, Kaohsiung 807, Taiwan; 15Department of Education and Research, Taipei City Hospital, Taipei 103, Taiwan

**Keywords:** polyetheretherketone crown, aging time, TiO_2_ content, fracture load

## Abstract

Polyetheretherketone (PEEK) is an emerging thermoplastic polymer with good mechanical properties and an elastic modulus similar to that of alveolar bone. PEEK dental prostheses for computer-aided design/computer-aided manufacturing (CAD/CAM) systems on the market often have additives of titanium dioxide (TiO_2_) to strengthen their mechanical properties. However, the effects of combining aging, simulating a long-term intraoral environment, and TiO_2_ content on the fracture characteristics of PEEK dental prostheses have rarely been investigated. In this study, two types of commercially available PEEK blocks, containing 20% and 30% TiO_2_, were used to fabricate dental crowns by CAD/CAM systems and were aged for 5 and 10 h based on the ISO 13356 specifications. The compressive fracture load values of PEEK dental crowns were measured using a universal test machine. The morphology and crystallinity of the fracture surface were analyzed by scanning electron microscopy and an X-ray diffractometer, respectively. Statistical analysis was performed using the paired *t*-test (α = 0.05). Results showed no significant difference in the fracture load value of the test PEEK crowns with 20% and 30% TiO_2_ after 5 or 10 h of aging treatment; all test PEEK crowns have satisfactory fracture properties for clinical applications. Fracture surface analysis revealed that all test crowns fractured from the lingual side of the occlusal surface, with the fracture extending along the lingual sulcus to the lingual edge, showing a feather shape at the middle part of the fracture extension path and a coral shape at the end of the fracture. Crystalline analysis showed that PEEK crowns, regardless of aging time and TiO_2_ content, remained predominantly PEEK matrix and rutile phase TiO_2_. We would conclude that adding 20% or 30% TiO_2_ to PEEK crowns may have been sufficient to improve the fracture properties of PEEK crowns after 5 or 10 h of aging. Aging times below 10 h may still be safe for reducing the fracture properties of TiO_2_-containing PEEK crowns.

## 1. Introduction

The excellent mechanical properties, thermal properties, and chemical resistance of polyetheretherketone (PEEK) make it very suitable for industrial applications [1]. In recent years, PEEK has become the most common polymer in the family of polyaryletherketones (PAEKs) used in the biomedical industry, such as as a replacement for titanium implants in the orthopedic field [2]. PEEK has good biocompatibility and radiographic penetrability, is widely used in the field of orthopedics, and has recently crossed over into dental applications, such as crown/bridge, implant abutments, frameworks, and clasps for partial dentures [3]. Compared to metals and ceramics, PEEK has a lower elastic modulus (approximately 3–4 GPa) similar to that of alveolar bone (approximately 11.5 GPa) [4], thus overcoming the risk of stress shielding effects and serving as an alternative to titanium and ceramic implants. In general, PEEK dental prostheses can provide a more comfortable feeling for patients during the chewing process. Moreover, tooth-color materials are becoming more popular due to higher aesthetic demands. This is one of the main reasons for the introduction of PEEK into dentistry, as it is more aesthetically pleasing than metal. A study by Beuer et al. [5] showed that PEEK has higher fracture resistance than ceramics. Therefore, it is important to investigate the mechanical properties of PEEK prostheses in an intraoral environment for long-term applications.

Although PEEK has the abovementioned excellent properties, its compressive strength is lower than that of cortical and alveolar bone. Adding titanium dioxide (TiO_2_) particles into the PEEK matrix, in addition to acting as a white dye (pigment) to adjust the color of PEEK, can improve the hardness of PEEK [6]. The bending modulus of PEEK increases with the addition of TiO_2_ [7]. PEEK containing 20% TiO_2_ has a significantly higher compressive strength than PEEK containing 10% TiO_2_ [8]. In addition, a study found that PEEK containing 10% TiO_2_ has a significantly higher elastic modulus but a significantly lower compressive strength when PEEK samples are immersed in Ringer’s solution at room temperature for 12 w. There are several studies on the physical and mechanical properties of TiO_2_-free PEEK used in dental applications [9,10,11], and the properties of TiO_2_-reinforced PEEK are not well understood. Previous studies showed that aging treatment can affect the color change and increase the surface roughness of 20% TiO_2_-containing PEEK [12]. However, it is not clear how the intraoral environments (e.g., temperature and moisture) affect the mechanical properties of PEEK dental prostheses, particularly TiO_2_-containing ones, over time. Therefore, the investigation of the role of aging time on the mechanical properties of PEEK with different TiO_2_ contents is clinically necessary.

We hypothesize that prolonged exposure to intraoral environments may affect the mechanical strength of dental crowns. Therefore, the corresponding experiments in this study were designed to simulate the effects of long-term intraoral environments by using the accelerated aging treatment noted in the ISO 13356 specifications. Our previous study found that compared to TiO_2_-free PEEK crowns, the addition of TiO_2_ increases the fracture load of PEEK crowns under non-aging treatment conditions, while the fracture load of the TiO_2_-containing PEEK crowns decreases after 5 h of aging treatment, in contrast to the TiO_2_-free PEEK crowns [13]. Our previous study showed that the aging treatment has an adverse effect on the fracture load of the PEEK crowns, depending on the presence of TiO_2_. However, the impact of a longer aging time on the fracture load of the TiO_2_-containing PEEK crowns may have more clinical implications. Although the effects of combining the aging time and TiO_2_ content on the fracture load of the TiO_2_-containing PEEK dental crowns are of clinical importance, no detailed information is available in the literature. This present study would like to confirm whether increasing TiO_2_ content in PEEK could have a more detrimental effect on fracture load at different aging times. Therefore, this study aimed to investigate the combined effects of aging time (5 and 10 h) and TiO_2_ content (20% and 30%) on the fracture characteristics of PEEK dental crowns.

## 2. Materials and Methods

### 2.1. PEEK Crown Preparation

Two types of commercial biocompatible high-performance polymer (BioHPP®) PEEK blocks (Ø 98.5 mm; thickness 20 mm) from the same manufacturer (bredent group GmbH & Co. KG, Senden, Germany) were used: 20% TiO_2_-containing PEEK (breCAM.BioHPP^®^ bredent white, designated as BreW) and 30% TiO_2_-containing PEEK (breCAM.BioHPP^®^ dentin-shade 2: Bredent A2, designated as BreA). These two types of PEEK blocks are on the dental market in the form of milling (namely for computer-aided design/computer-aided manufacturing (CAD/CAM) fabrication). Considering the mechanical properties in dental clinical applications, PEEK blocks containing 20% and 30% TiO_2_ are the most commonly used prosthesis materials in digital dental fabrication. To avoid differences in the mechanical properties of different brands of PEEK blocks (even though they have the same TiO_2_ content), the commercial 20% and 30% TiO_2_-containing dental PEEK blocks from the same brand were selected for this study.

The test BreW and BreA PEEK blocks from the same brand have similar physical, chemical, and mechanical properties, differing only in TiO_2_ content and color, with data provided by the manufacturer as follows (https://bredent-group.com/wp-content/uploads/2019/12/breCAM_consumables_2018_000500GB_20190211_low.pdf, accessed on 13 June 2023): elastic modulus ≥ 4.2 GPa, flexural strength ≥ 160 MPa, water solubility ≤ 0.3 μg/mm^3^, water absorption ≤ 6.5 μg/mm^3^, wearing time ~ permanent, and residual monomer = 0%. These two types of BioHPP^®^ PEEK blocks have a glass transition temperature of around 151 °C [14].

The artificial abutment (LR 62, Nissin Dental Product Inc., Kyoto, Japan) (Figure 1) was selected from the first molar of the lower-right mandible (#46), which has an elastic modulus similar to that of a natural tooth [4], and was therefore chosen to simulate a natural tooth abutment. The artificial tooth abutment used for the PEEK crown was prepared according to the clinically recommended finishing line (margin) of the tooth preparation thickness of 0.50 mm and occlusal surface thickness of 1.0 mm. Each artificial tooth abutment was scanned directly with a scanner (inEos X5, Dentsply Sirona, Charlotte, NC, USA), and a CAD system (built-in software, 3Shape, Copenhagen, Denmark) was used to create a consistent crown shape. The PEEK crowns of the first molar of the lower-right mandible were then fabricated by a CAM system (MC X5, Dentsply Sirona, Charlotte, NC, USA).

### 2.2. Aging Treatment of Crown

20% TiO_2_-containing PEEK (BreW) and 30% TiO_2_-containing PEEK (BreA) dental crowns were treated for two different aging times, including the aging 5 h group (designated as A5) and the aging 10 h group (designated as A10). The aging treatment was based on ISO 13356 specifications, wherein the materials were autoclaved and subjected to 134 °C and 2 bar for 5 h to simulate a long-term intraoral environment. At a 37 °C intraoral environment, five hours of accelerated aging at 134 °C and 2 bar correspond to 15–20 y [15]. Therefore, a 10-h aging treatment was expected to simulate 30–40 y of PEEK crown in the intraoral environment, which is enough for long-term clinical applications.

### 2.3. Bonding PEEK Crown to Artificial Abutment Tooth

The crown samples were cleaned by shaking them in 95% alcohol and then dried with air. A dental blasting machine was used to blast the inside of the crown with 120 µm aluminum oxide particles at 2 bar pressure for 10 s, with a distance of 10 cm between the nozzle and the sample. The sample was then cleaned with distilled water and blown dry with pressurized and filtered gas from a dental air gun. Finally, a medical-grade adhesive (EA M-31 CL, Loctite, Rocky Hill, CT, USA) was applied to the interior of the crowns and the surfaces of the artificial abutments, and the crowns were placed on the artificial abutments and pressed for 1 min to remove the excess adhesive from the edges of the crowns. The abutments cemented with PEEK crowns were embedded in the epoxy resin (EpoFix Kit, Struers, Rødovre, Denmark) (Figure 2) before performing the subsequent compressive tests, as described below.

### 2.4. Fracture Load Measurement

All crown samples adhered to the artificial abutments were subjected to compressive tests, performed using a universal test machine (Bionix^®^-858, MTS, Eden Prairie, MN, USA) with a loading rate of 1 mm/min (Figure 3). The loading head was made of 430 stainless steel with a half-round diameter of 3 mm, and a 0.5 mm thick co-polyester spacer (Imprelon^®^ Spd, Scheu-Dental GmbH, Iserlohn, Germany) was placed between the loading head and the PEEK crown in order to apply a uniform force to the surface of the PEEK crown. The moment of fracture of the PEEK crown was recorded as the fracture load (N), and all fractured PEEK fragments were collected and analyzed as mentioned in the following.

### 2.5. Morphology Observation and Crystalline Analysis of Fracture Surface

The macroscopic fracture appearance of the fractured PEEK crowns was observed visually. To observe the microscopic fracture pattern of PEEK crowns, the non-conductive fracture surface was analyzed using a scanning electron microscope (SEM) (JSM-6500F, JEOL, Tokyo, Japan) with secondary electron image mode and 15 kV acceleration voltage after being coated with a thin (~10 nm) conductive platinum film. Meanwhile, an energy dispersive spectrometer (EDS) (Ultim Max 100, Oxford Instruments, Abingdon-on-Thames, UK) with mapping analysis mode was used to analyze the elemental distribution of titanium (Ti), oxygen (O), and carbon (C) on the fracture surface to confirm the dispersion of TiO_2_ particles in the PEEK matrix. Furthermore, the crystallinity of the fracture surface was analyzed using a high-intensity X-ray micro-area diffractometer (D8 Discover, Brucker, Billerica, MA, USA) with beam sizes down to 180 × 180 μm^2^ to investigate the crystalline structure of test PEEK crowns. The measured 2θ ranged from 10 to 80°, with a scanning rate of approximately 1° per minute.

### 2.6. Statistical Analysis

For fracture load measurement, the number of samples was 5 for each test group, and the results were expressed as the mean  ±  standard deviation (SD). A paired *t*-test using IBM^®^ SPSS^®^ Statistics software (version 21) was used to determine the level of significance, and *p*  <  0.05 was considered statistically significant. The sample size was calculated using G*Power software (version 3.1.9.7).

## 3. Results

### 3.1. Fracture Load: Effects of Aging Time and TiO_2_ Content

Figure 4 presents the fracture loads of the test PEEK crowns (BreW and BreA) after 5 and 10 h of aging treatments. The corresponding statistical analysis results using the paired *t*-test are listed in Table 1. No significant difference in fracture load was observed among the test groups (*p* > 0.05).

For a 20% TiO_2_-containing PEEK crown (BreW), the fracture load values were in the following order: BreW-A5 group (8972 ± 807 N) > BreW-A10 group (8794 ± 1140 N) (Figure 4). However, the paired *t*-test analysis results showed no significant difference in fracture load between the BreW-A5 and BreW-A10 groups. For a 30% TiO_2_-containing PEEK crown (BreA), the fracture load values were in the order: BreA-A10 group (7950 ± 404 N) > BreA-A5 group (7773 ± 1523 N) (Figure 4). Similar results showing that aging time had no significant influence on the fracture load of BreW crowns were also observed for BreA crowns.

As shown in Figure 4, for 5 or 10 h of aging treatments, the paired *t*-test analysis results showed that there was no significant difference in fracture load between the 20% TiO_2_-containing BreW and 30% TiO_2_-containing BreA groups under the same aging time, even though the mean fracture load values of BreW groups were higher than those of BreA groups.

### 3.2. Morphology of Fracture Surface

The fracture surface morphology of the PEEK crowns after compressive tests was observed visually (Figure 5) and by SEM (Figure 6). For both BreW and BreA crowns, regardless of the aging time and TiO_2_ content, it was observed visually that the fracture extended from the occlusal surface of the crown along the lingual sulcus to the edge of the crown and finally fractured. Under higher magnification conditions (Figure 6), the fracture surface could be microscopically divided into three parts: the beginning of the fracture, the middle extension of the fracture, and the end of the fracture.

Figure 7 shows a SEM micrograph of the fractured surface of BreA-A5 after grinding and its corresponding EDS mapping analysis of Ti, O, and C elements. The brightly colored spots indicated relatively high elemental content. The results of the EDS analysis showed that the Ti and O elements were uniformly dispersed in the 30% TiO_2_-containing PEEK matrix.

### 3.3. Crystallinity of Fracture Surface

Figure 8 presents the XRD spectra of the fracture surfaces of BreW and BreA crowns. Regardless of aging time, all test crowns showed a predominantly PEEK crystalline structure. Additionally, the peaks at 27.5°, 36°, 41°, 54.5°, and 56.5° were also observed for all test crowns, indicating the presence of rutile phase TiO_2_ crystalline [16]. In addition, some weak peaks at 26°, 29°, and 43° were only observed in the BreA crowns, indicating the presence of trace amounts of BaSO_4_ [17,18].

## 4. Discussion

### 4.1. Effect of Aging Time on Fracture Load

The mechanical properties of PEEK materials without aging treatments have been widely investigated [6,7,8,9,10,11]. The aging treatment can affect the color change and increase the surface roughness of 20% TiO_2_-containing PEEK [12]. Therefore, there is a clinical need to investigate the role of aging time, simulating long-term clinical application, on the mechanical properties of CAD/CAM-fabricated TiO_2_-containing PEEK crowns. The aging process used in this study was based on the ISO 13356 specification, which is commonly used for dental zirconia, and the test samples were aged in an autoclave at a temperature of 134 °C and a pressure of 2 bar for 5 h. It is estimated that 5 h of aging at 134 °C is equivalent to 15–20 y in an intraoral environment [15]. PEEK material gains more popularity, especially in dentistry, due to its good mechanical strength and aesthetic properties. However, as an emerging polymer material in dentistry, PEEK does not yet have an aging-related ISO specification to follow, so the ISO 13356 specification used for dental zirconia was used as a reference in this study. Previous studies have also applied the same hydrothermal aging treatment (134 °C/2 bar/5 h) for PEEK to investigate the effect of aging on the mechanical properties of commercial dental PEEK materials [19].

According to the ISO specifications, the aging treatment can be detrimental to the mechanical properties of dental zirconia. Thus, we expect that the aging treatment will result in a decrease in the fracture load value of PEEK. In this study, the TiO_2_-containing BreW and BreA crowns showed no significant difference in the fracture load between 5 and 10 h of aging (Figure 4). Our present study found that even though there was no significant difference between the two TiO_2_-containing PEEK groups, the fracture load of 30% TiO_2_-containing PEEK was slightly lower than that of 20% TiO_2_-containing PEEK. This could be due to slightly different compositions. However, this corresponded to our previous study, which found that under aging conditions, PEEK without TiO_2_ has a higher fracture load than TiO_2_-containing PEEK. The present results implied that more TiO_2_ content might not improve mechanical properties or detrimentally affect the mechanical properties of PEEK after aging treatment. This needs further investigation to confirm the above speculation.

A previous study found a significant increase in the flexural strength of unreinforced PEEK (without any additives) after annealing treatment at 250 °C for 4 h [20]. This is similar to our previous findings that TiO_2_-free PEEK increases its fracture load after 5 h of aging [13]. In the present study, the fracture load of the PEEK crown containing either 20% or 30% TiO_2_ was not affected by increasing the aging time from 5 to 10 h. We presumed that a critical aging time to decrease the fracture load of the TiO_2_-containing PEEK crowns might be above 10 h; thus, no differences were observed in the fracture load of the test PEEK crowns under an aging time below 10 h. Therefore, the underlying mechanism of the effect of aging time (>10 h) on the fracture properties of the TiO_2_-conating PEEK crowns still needs to be further investigated.

### 4.2. Effect of TiO_2_ Content on Fracture Load

Previous studies have shown that the elastic modulus, hardness [6], and bending modulus [7] of PEEK specimens increase with increasing TiO_2_ content. The cylindrical PEEK containing 20% TiO_2_ has a significantly higher elastic modulus compared to that containing 10% TiO_2_ [8]. Therefore, taking into account the TiO_2_ contents commonly used in clinical practice, there is a clinical need to study the effect of TiO_2_ content on the mechanical properties of CAD/CAM-fabricated PEEK crowns with aging treatment. In this study, the statistical analysis result showed that the amount of TiO_2_ content (20% or 30%) did not affect the fracture load values of aging-treated PEEK crowns (Figure 4), regardless of the aging time. This differs from the literature mentioned above, presumably due to the difference in the shape of the test specimens and the presence of aging treatment. In this study, the first molar crown on the lower-right mandible was prepared, whereas in the previous literature, standard specimens in the form of bars or cylinders were prepared. In addition, the choice of artificial abutment material and cement used in preparing the crown samples may have played a role. The 20% TiO_2_ content may have reached the minimum TiO_2_ content requirement for improving fracture characteristics. This result, however, needs to be further investigated in the future.

### 4.3. Morphology of Fracture Surface

We visually observed the fracture surface of each group of PEEK crowns, starting from the lingual side of the occlusal surface of the crown and extending along the lingual groove to the margin of the crown (Figure 5). Microscopically, the fracture surface could be divided into three parts: the fracture’s beginning, the fracture’s middle extension, and the fracture’s end (Figure 6). The beginning and middle extensions of the fracture surface had a flat and smooth morphology, similar to the result observed in Bragaglia et al.’s study [21]. Under higher magnification conditions, we found that the middle crack extension of all test PEEK groups was mainly feather-shaped, and there was no significant difference in the feather shape as the aging time increased. The end fracture morphology was mainly coral-shaped with granules.

Referring to Bragaglia et al.’s study [21], the authors found that the crack growth zone in the mid-extension part of the fracture surface of PEEK samples after the tensile test is also microscopically feathery (called parabolic fracture/finger-like structure), while the end fracture zone is also coral-like (called fast fracture region). The authors suggest that cracks in the PEEK matrix extend along the fragile non-crystalline structure between the spherical crystals (spherulites), thus forming a feathery shape. The more TiO_2_ particles are added, the smaller the spherical crystals will be and the finer the non-crystalline structure. This explained the fact shown in Figure 6 that the 30% TiO_2_-containing BreA crowns appeared to have a more distinct feather-like shape on the fracture surface than the 20% TiO_2_-containing BreW crowns. However, the aging time did not significantly affect the microscopical fracture morphology.

We observed that the feather-shaped structure was covered with a particle-like material in the middle fracture extension, which was presumed to be TiO_2_ particles. We estimated the size of the particles to be about 100~200 nm based on SEM micrographs (Figure 6), which could not be quantitatively confirmed by energy dispersive spectroscopy (EDS) analysis accompanying the SEM instrument because the electron beam size of EDS point analysis is in the range of a few thousand nm. Therefore, the EDS mapping analysis was used to qualitatively confirm the uniform distribution of TiO_2_ particles in the BreA-A5 matrix (Figure 7). Similar results were also observed in other test groups, i.e., TiO_2_ particles were uniformly distributed in the PEEK matrix regardless of the aging time and TiO_2_ content.

### 4.4. Crystallinity of Fracture Surface

The crystallinity of the fracture surface of all test PEEK crowns was analyzed by XRD (Figure 8). Results showed three weak peaks around 20° and one weak peak around 30° for each group of PEEK, regardless of the aging time and TiO_2_ content. These peaks correspond to the crystalline nature of PEEK [22]. In addition, significant peaks were observed at approximately 27.5°, 36°, 41°, 54.5°, and 56.5°, which represented the rutile phase of TiO_2_ [10]. The crystalline structure of all test PEEK crowns was still dominated by the PEEK matrix and rutile phase TiO_2_, regardless of the aging time and TiO_2_ content.

TiO_2_ exists in three crystalline phases: rutile, anatase, and brookite. Rutile is the most stable crystalline state in the atmosphere, and each of the three crystalline phases has its own stable particle size: the stable particle size of the rutile phase TiO_2_ is larger than 35 nm, while the most stable particle size of the anatase and brookite phases TiO_2_ is smaller than 35 nm [23]. The size of TiO_2_ particles has been estimated to be 100–200 nm based on the SEM images (Figure 6), and it meets the condition of stable rutile phase TiO_2_ size. Furthermore, according to the report from the manufacturer of BioHPP^®^ PEEK used in this study, the grain size of the TiO_2_ particles in the PEEK matrix is about 300–500 nm (https://www.bredent.co.uk/wp-content/uploads/2017/02/BioHPP-2013.pdf, accessed on 13 June 2023). Therefore, the rutile phase TiO_2_ was observed in the XRD spectra (Figure 8) of all test PEEK crowns. Regardless of aging time, the presence of 20% rutile phase TiO_2_ in BreW resulted in a slightly higher fracture load than that of 30% rutile phase TiO_2_ in BreA, which should be further investigated. Moreover, this corresponded to our previous study [13] showing that under aging treatment, the TiO_2_-free PEEK has a higher fracture load than the TiO_2_-containing PEEK. Thus, this present finding displayed that even though the fracture load of the two PEEK groups was not statistically significant difference, the 30% TiO_2_-containing PEEK had a slightly lower fracture load than the 20% TiO_2_-containing one. For a detailed study of the crystallinity of PEEK, refer to previous literature [24].

In addition, the crystallographic analysis of the BreA group revealed sharp peaks between 25° and 45° (blue inverted triangles for BreA-A5 and BreA-A10 in Figure 8), which, in comparison with the previous studies [17,18], are presumed to be the crystalline structure of BaSO_4_. Therefore, in addition to the PEEK matrix and rutile phase TiO_2_, the BreA groups were confirmed to contain small amounts of BaSO_4_. It is known that BaSO_4_ has been used as an additive to enhance the mechanical strength of PEEK, but it is not as effective as TiO_2_. Therefore, most PEEK on the market still uses TiO_2_ as the main additive [8]. One possible purpose of adding BaSO_4_ to commercial PEEK is to enhance the visibility of the polymer in medical X-ray and magnetic resonance imaging (MRI) tests [25].

To estimate whether the glass transition temperature (Tg) was different among the test groups, differential scanning calorimetry (DSC) (DSC 4000, PerkinElmer, Shelton, USA) analysis was performed on the test PEEK groups between room temperature and 200 °C. Results showed no significant difference in Tg among the test groups (Appendix A). This implied that the aging treatment and TiO_2_ content did not have a significant effect on the Tg of commercial dental PEEK materials used in this study.

### 4.5. Clinical Significance

Barba et al. studied the tensile strength of additive-free PEEK at different temperatures [26]. The authors found that a higher temperature, particularly one equal to or higher than Tg, softens PEEK and increases its ductility. Most commercial PEEK materials have a Tg of approximately 143 to 145 °C [10,27]. In this study, the BioHPP^®^ PEEK crowns used were aged at a temperature (134 °C) below their Tg (approximately 151 °C [14]). The aging time (5 or 10 h) did not statistically affect the fracture load and surface morphology of PEEK crowns, regardless of the TiO_2_ content.

It is known that adding TiO_2_ to PEEK increases the mechanical properties of TiO_2_-free PEEK [5,6,7,8]. In this study, the TiO_2_ content (either 20% or 30%) did not statistically affect the fracture load of PEEK crowns (*p* > 0.05), regardless of the aging time, although the PEEK crowns with 30% TiO_2_ showed a more distinct feather-like shape on the fracture surface. This means that a 20% TiO_2_ addition may already meet the minimum requirements for improving the mechanical properties of PEEK crowns; aging times below 10 h may still be safe for degrading the mechanical properties of TiO_2_-containing PEEK crowns. The results of this study showed that regardless of the aging time and TiO_2_ content, the fracture loads (7800–9000 N) of the test PEEK crowns in all groups were much higher than the average human maximum mastication force (approximately 163~539 N) [28,29,30] and those (approximately 2500–3000 N) obtained by Shirasaki et al. [31]. Note that the *p*-value (0.059) for the paired *t*-test analysis of BreW-A5 vs. BreA-A10 in Table 1 was a little greater than 0.05, namely marginally significant. This may imply that there could still be some interaction between the BreW-A5 and BreA-A10 groups. This, however, needs further investigation using larger sample sizes.

Although our research was not an in vivo or clinical simulation study, we attempted to create environments that could be translated into clinical situations. We performed aging conditions to simulate the long-term oral environment and applied loading to simulate masticatory force. Considering the clinical situation, the test materials used were clinical CAD/CAM PEEK blocks, and the crown sample was identical to the first molar of the lower-right mandible (#46). However, this was a preliminary study before establishing a more complex clinical testing environment for PEEK crowns.

Until now, no adequate data to support the PEEK dental prostheses’ long-term endurance has been available in the literature [32]. The fracture load values of both PEEK groups did not significantly change after 5 or 10 h of aging treatment (*p* > 0.05). This showed that the TiO_2_-containing PEEK crowns had excellent resistance to chewing fractures after simulating a long-term intraoral environment. In clinical dental applications, the commercial TiO_2_-containing PEEK can be an alternative restorative material for patients with metal allergies or those with more aesthetic considerations. However, this was only the result of compression tests in this study, and further dynamic cyclic loading tests should be performed to simulate the dynamic masticatory loading environment of the oral cavity. Additionally, TiO_2_ could improve the mechanical properties of PEEK, but it may have potential adverse effects under aging conditions. In future studies, the mechanism by which the fracture load of TiO_2_-containing PEEK is reduced after aging treatment needs to be investigated. Moreover, the effect of a longer aging time (>10 h) can also be further explored.

Dental CAD/CAM PEEK blocks can be applied for dental prostheses, including abutments, crowns/bridges, implants, etc. In our study, we considered using the clinical crown-shaped PEEK for compression tests. We should also consider making the PEEK implant-shape standard samples for compression (or bending) tests and doing the dynamic cyclic loading tests based on ISO 14801 in the future.

Considering the limitations of this study and future clinical applications, larger sample sizes, longer aging periods, and cyclic loading situations are suggested in the perspective research. Furthermore, investigating the effects of other additives (e.g., carbon fiber) and testing the PEEK crowns in vivo are also recommended. Currently, a study on the impact of fatigue treatment on the fracture load of PEEK crowns is underway. Preliminary results (based on 3 samples per test group) showed that after 500,000 cycles of compressive loading (frequency: 2 Hz; load: 30–300 N) in an artificial saliva environment (pH 5.4; 37 °C) [33], the average fracture loads of the fatigue-treated PEEK crowns were approximately 8100 N for BreW (~900 N lower vs. BreW-A5 in Figure 4) and 7400 N for BreA (~400 N lower vs. BreA-A5 in Figure 4). The effect of fatigue treatment on the fracture characteristics of TiO_2_-containing PEEK crowns after aging treatments will continue to be evaluated in the future.

## 5. Conclusions

Based on the results obtained in this study under the experimental limitation, there were no statistically significant differences in the fracture load of the test commercial PEEK crowns, regardless of the aging time (5 or 10 h) and TiO_2_ content (20 or 30%). The fracture loads obtained for all test PEEK crowns were much higher than the average human occlusal force. This implied that either a 20% or 30% TiO_2_ addition might already meet the minimum requirements for improving the mechanical properties of PEEK crowns. Aging times below 10 h might still be safe for degrading the mechanical properties of TiO_2_-containing PEEK crowns. Therefore, the commercial TiO_2_-containing PEEK crowns investigated in this study are suitable for clinical dental applications. Further investigation of the effects of combined dynamic chewing force on the fracture properties of TiO_2_-containing PEEK crowns has been suggested.

## Figures and Tables

**Figure 1 polymers-15-02720-f001:**
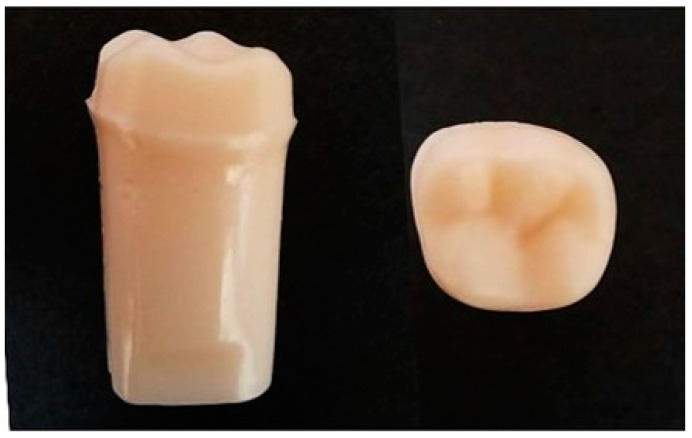
Artificial abutment used in this study (left: side view; right: top view).

**Figure 2 polymers-15-02720-f002:**
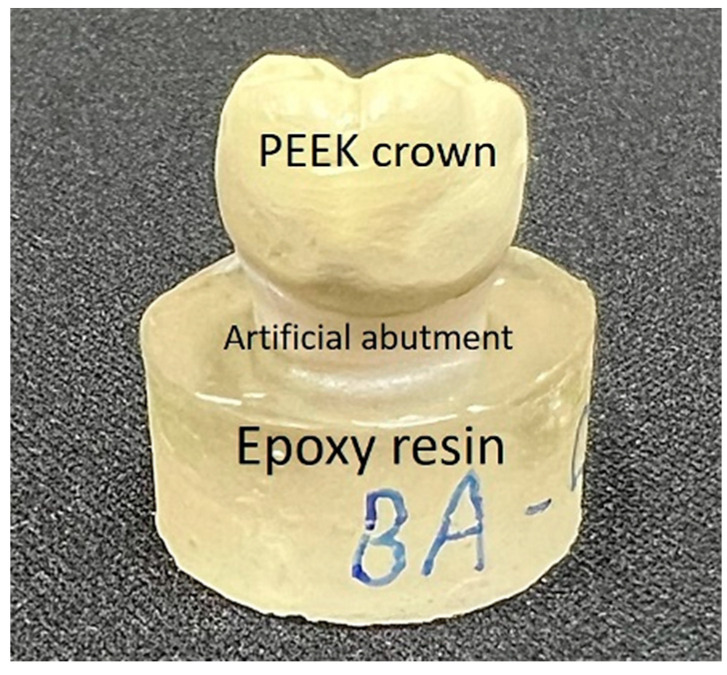
Test samples for the compressive test (top: PEEK crown; middle: artificial abutment; bottom: epoxy resin).

**Figure 3 polymers-15-02720-f003:**
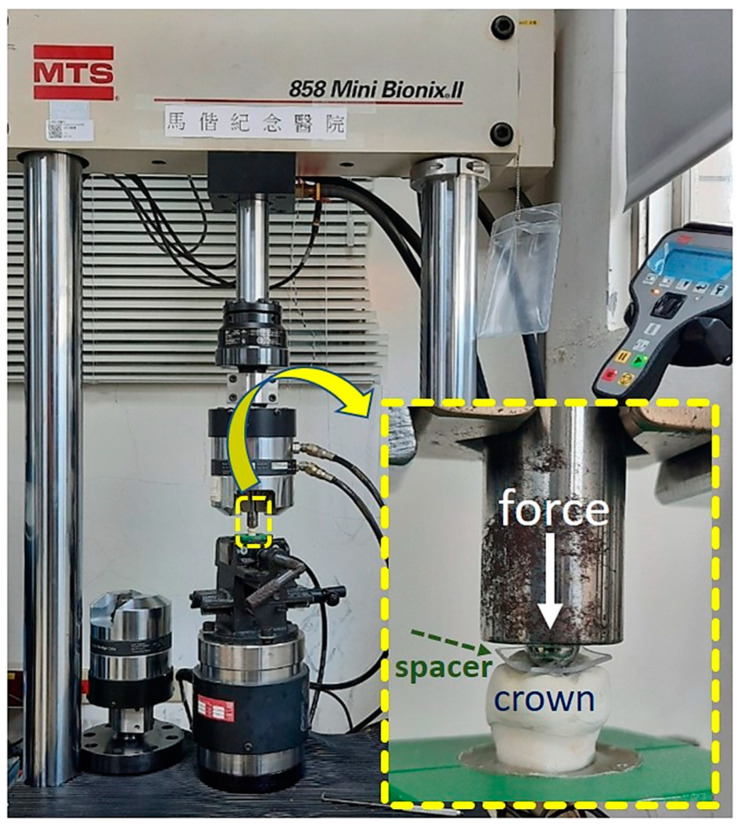
The layout of compressive test (lower-right inset: higher magnification of test crown being loaded).

**Figure 4 polymers-15-02720-f004:**
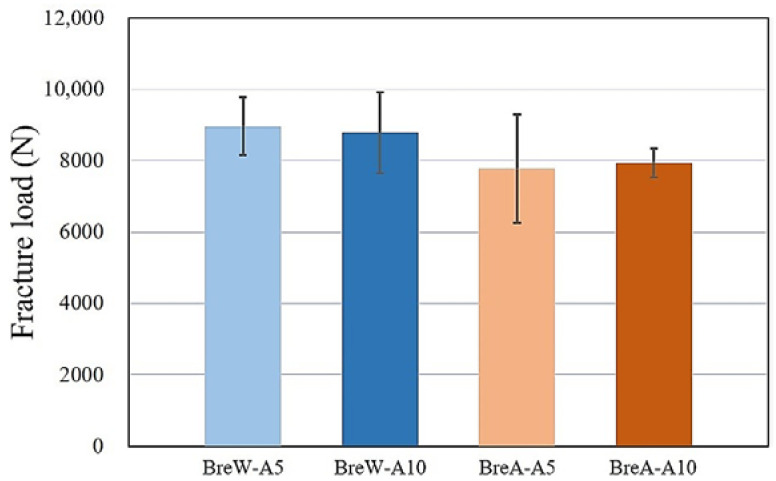
Fracture load (N) vs. test PEEK crowns (BreW and BreA) after 5 and 10 h of aging treatments: BreW-A5, BreW-A10, BreA-A5, and BreA-A10. No significant difference in fracture load was observed among the test groups after the paired *t*-test analysis (*p* > 0.05).

**Figure 5 polymers-15-02720-f005:**
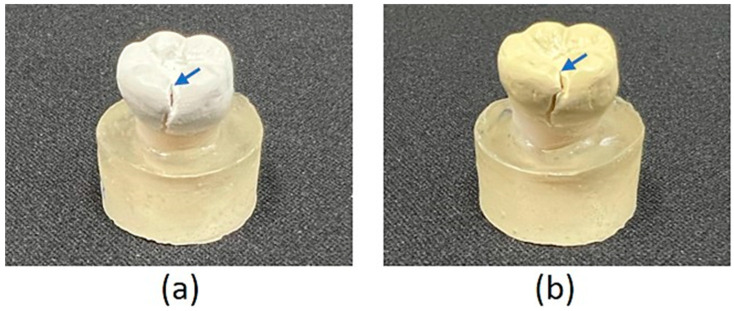
Visual observation of fracture surface morphology of test PEEK crowns: (**a**) BreW-A5; (**b**) BreA-A5 (arrows representing the cracking initiation), showing the fracture extended from the occlusal surface of the crown (as indicated by the arrow) along the lingual sulcus to the edge of the crown and finally ruptured.

**Figure 6 polymers-15-02720-f006:**
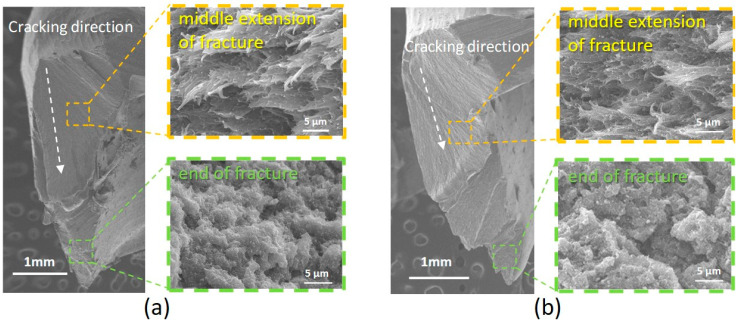
SEM observation of fracture surface morphology of test PEEK crowns: (**a**) BreW-A5; (**b**) BreA-A5, showing the beginning of the fracture, the middle extension of the fracture, and the end of the fracture.

**Figure 7 polymers-15-02720-f007:**
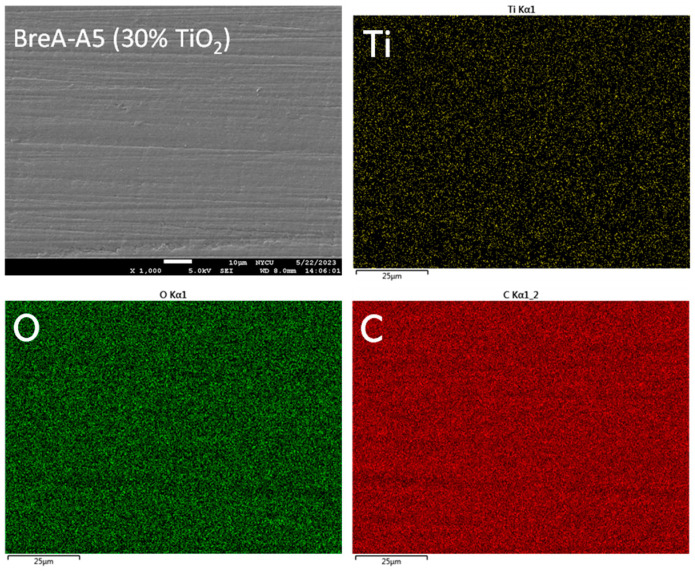
SEM micrograph of the fracture surface of BreA-A5 after grinding (upper-left) and the corresponding EDS mapping analysis of the Ti, O, and C elements, showing the Ti and O elements were evenly distributed in the TiO_2_-containing PEEK matrix.

**Figure 8 polymers-15-02720-f008:**
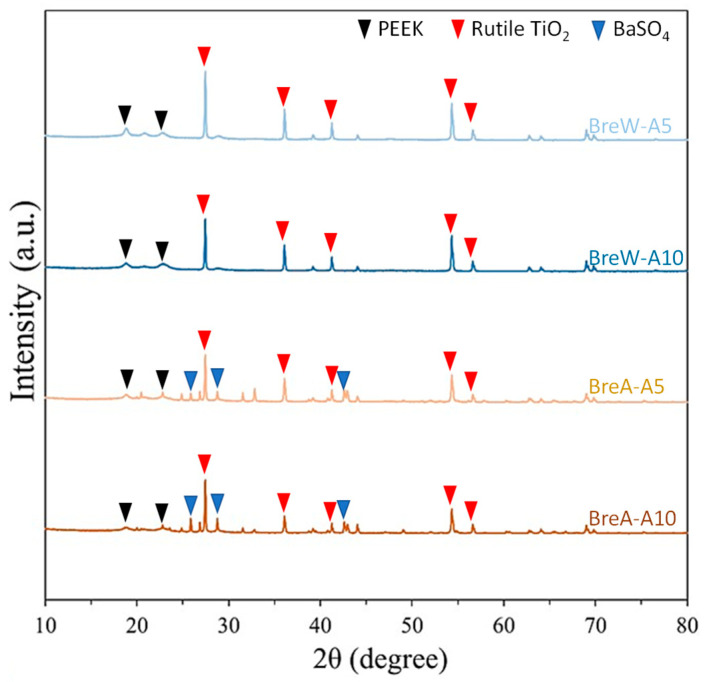
Crystallinity of the fracture surface of test PEEK crowns: BreW-A5, BreW-A10, BreA-A5, and BreA-A10, showing rutile TiO_2_ was present in all test PEEK crowns, while BaSO4 was only present in the BreA groups.

**Table 1 polymers-15-02720-t001:** Statistical analysis results for fracture load (N) shown in Figure 4.

	Mean	SD	95% Confidence Interval		Degree of Freedom	*p*-Value
			Lower	Upper		
BreW-A5: 1	8972.134	361.339091	7968.895850	9975.372150		
BreW-A10: 2	8794.060	509.953859	7378.201103	10,209.918897		
BreA-A5: 3	7773.816	681.394623	5881.961234	9665.670766		
BreA-A10: 4	7950.882	180.744110	7449.055901	8452.708099		
1 vs. 2					4	0.775965
1 vs. 3					4	0.152252
1 vs. 4					4	0.059349
2 vs. 3					4	0.323025
2 vs. 4					4	0.143422
3 vs. 4					4	0.820478

*p* < 0.05: significant difference.

## Data Availability

The data presented in this study are available on request from the corresponding author upon reasonable request.

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
