# Peer review of "Fracture Characteristics of Commercial PEEK Dental Crowns: Combining the Effects of Aging Time and TiO2 Content"

_polymers, 2023, doi:10.3390/polym15122720_

Round 1

Reviewer 1 Report

This manuscript is a similar work as compared to the previous published work by the same authors as reference 8. The reference 8 compared the effect of loading and aging on the additive-free PEEK crown and PEEK crown with 20% TiO2 particles. The current work compared the effect of loading and aging on the PEEK crown with 20% TiO2 particles and 30% TiO2 particles. The application importance of this work is evident.

1. However, the design of the experiments could be improved, probably PEEK with different content of TiO2, such as 0%, 10%, 20%, 30%, and at least three aging hours would be more systematic, and the design of the experiments would be sounder, rather than just pick two concentrations and only compare only these two.

2. Introduction part, “Compared to metals and ceramics, PEEK has a lower elasticity coefficient (approximately 3-4 GPa)”. The “elasticity coefficient” should be “elasticity modulus”

3. In ref. 8, EDS test of the fracture surfaces revealed the distribution of TiO2, actually that is very interesting results and if that test could also be done on the 30% TiO2, that’ll be better.

4. Part 4.2. “Previous studies have shown that the bending coefficient and hardness of strip PEEK specimens increase with the addition of TiO2 [5,6].” There was no bending coefficient mentioned in these two references, reference 6 studied bending deflection, however, there were no TiO2 additives in reference 6. Please correct this problem.

Pretty good with some minor errors.

Grammar errors: 1. in addition to as a white dye, 2.2. dental crowns were performed two different aging times.

Reviewer 2 Report

Overall, this study provides valuable insights into the fracture characteristics of commercially available PEEK dental prostheses. However, there are several areas that need to be addressed in order to strengthen the paper and make it suitable for publication:

1.     The introduction could be improved by providing more information about the clinical relevance of the study. Why is it important to investigate the effects of aging and TiO2 content on the fracture characteristics of PEEK dental prostheses? What are the potential implications for clinical practice?

2.     The methodology section needs to be expanded to provide more detailed information about the experimental design, including the sample size, fabrication method, and testing conditions. It would also be useful to include information about the selection criteria for the PEEK blocks and the rationale for choosing the TiO2 content levels.

3.     The results section could be improved by providing more detailed information about the statistical analysis, including the significance level and confidence intervals. In addition, the findings could be better presented with the use of figures or tables to aid the reader's understanding.

4.     The authors need to provide a more thorough review of the existing literature on the topic and cite relevant studies that have investigated the mechanical properties of PEEK dental prostheses under intraoral environments. Additionally, the authors should discuss the limitations of previous studies and how their study contributes to the existing knowledge.

5.     It is unclear why a platinum coating was applied to the fracture surface of the PEEK crowns. This information should be clarified in the methodology section.

6.     The SEM and XRD analyses should be described in more detail, including the specific parameters used in the analyses.

7.     The results section should include a more detailed explanation of the statistical analysis, including the degrees of freedom and p-values for the paired t-tests.

8.     The figure legends should be revised to provide more detailed descriptions of the data presented in each figure.

9.     The authors should provide a more detailed description of the statistical analyses performed, including the type of test used (e.g., ANOVA), the alpha level used for significance, and the post-hoc tests performed.

10.  The authors should clarify the reasoning behind the choice of aging time used in the study and provide a more detailed description of the aging process used for PEEK.

11.  The authors should provide a more detailed explanation of the results, particularly in relation to the literature cited.

12.  The authors should provide more details on the statistical analysis used in the study. For example, they should state the level of significance used and provide the statistical values for the results.

13.  The authors should provide more discussion on the significance of their findings. For example, they should discuss the clinical implications of their results, such as the potential use of TiO2-containing PEEK crowns in dental prostheses and the need for further investigation of the effects of combined dynamic chewing force on the fracture properties of TiO2-containing PEEK crowns.

14.  The discussion section should be expanded to provide a more thorough interpretation of the results and their implications for future research. Specifically, the authors should discuss the potential reasons why TiO2 content did not significantly affect fracture load and whether this finding has any practical implications for the use of PEEK crowns in dental applications.

15.  The authors need to be more cautious in interpreting their results. While the study showed no significant difference in fracture load values between the two types of PEEK crowns, it is important to note that the sample size may have been too small to detect a significant difference. Additionally, the study only investigated the fracture characteristics of PEEK crowns in vitro, and it is unclear how these findings translate to clinical settings.

16.  The limitations of the study should be discussed, including the need for larger sample sizes and longer aging periods to better simulate clinical conditions. Additionally, suggestions for future research could be provided, such as investigating the effects of other additives or testing the PEEK crowns in vivo.

17.  The manuscript could benefit from improved language and formatting, including proper use of scientific terminology and organization of the sections. Additionally, the manuscript could be proofread for grammatical errors and typos.

18.  The literature review is insufficient. Kindly include the specific literatures such as: Current Nanoscience 18, no. 2 (2022): 203-216 https://doi.org/10.2174/1573413717666210216120741; Journal of Molecular Liquids 357 (2022): 119129 https://doi.org/10.1016/j.molliq.2022.119129

Minor editing of the English language required

Reviewer 3 Report

The paper can be accepted after minor revision. The following comments need to be addressed.

1.      Abstract: Provide a concise finding of what the article is about and the main conclusions that the authors have drawn from their research.

2.      Introduction: Problem statement and novelty of this study should be mentioned properly. This part should also be improved with up to dated reference citation.

3.      Materials and Method: Please provide materials name used in this study and their important properties such as chemical, mechanical and physical are required to mention in materials section.

4.      EEK Crown Preparation procedure is not clear.

5.      SEM test procedure is expected.

6.      Statistical Analysis: Please provide software name with version.

7.      Discussion: More discussion is required for the article's contributions to the field.

8.      Conclusion: Make recommendations for future research. Suggest possible avenues for further research that could build on the current study's findings or address any limitations or unanswered questions.

Minor English Language correction is required.

Reviewer 4 Report

The presented paper is focused on evaluating the influence of the heat aging process on the mechanical performance of TiO2-modified PEEK. In my opinion, the presented study lacks novelty since the use of modified PEEK materials is quite extensively described in the literature. 

Besides, I have a few comments regarding the methodology itself, which may be useful before re-submitting the text.

1. The heat aging standard ISO 13356 measurements are typically used for the evaluation of ceramic-based materials, like zirconia, while the glass transition region for PEEK starts at 130-140 C deg. Does long-term exposure increase the level of crystallinity of the matrix? In this context, it would be worthwhile to perform additional tests on samples before and after aging (thermal analysis like DSC or DMA).

2. Evaluation of properties based on implant-shaped sample compression tests should be the final stage of a series of tests for standardized samples. In my opinion, it should be preceded by a series of detailed comparative tests using standard testing methods like tensile or bending.

3. A large part of the research work on the durability of dental implants is devoted to resistance to dynamic loads in accordance with the ISO 14801 standard. The tests of this type would be a great addition to the presented results of the static tests.

Round 2

Reviewer 2 Report

The authors have revised the manuscript significantly, addressing all the comments and concerns raised during the initial review process. They are advised to add some relevant literature in the manuscript as: https://doi.org/10.2174/1573413717666210216120741, https://doi.org/10.3390/coatings12101459.

The authors have restructured the content to enhance the coherence and flow of ideas, ensuring that the key points are effectively conveyed. They have also incorporated additional experiments and analyses as suggested, which have strengthened their arguments and contributed valuable insights to the study.

Moreover, the authors have taken your technical comments into careful consideration. Based on the revisions made by the authors, I believe that the manuscript has been sufficiently improved and now meets the high standards of this journal, thus could be accepted for publication after inclusion of suggested articles.

Author Response

Thank you for your valuable comments.

Reviewer 4 Report

Unfortunately, I stand by my previous opinion. The revised version of the manuscript was only slightly corrected, most of the corrections relate to changes in text formatting and sentence order, so they cannot be considered valuable.

Author Response

Thank yo ufor your valuable comments. We have done out best to revise the manuscript accordingly.